# Student Cheating Detection in Higher Education by Implementing Machine Learning and LSTM Techniques

**DOI:** 10.3390/s23084149

**Published:** 2023-04-20

**Authors:** Waleed Alsabhan

**Affiliations:** College of Engineering, Al Faisal University, P.O. Box 50927, Riyadh 11533, Saudi Arabia; walsabhan@alfaisal.edu

**Keywords:** student cheating detection, machine-learning, exploratory data analysis, online examination, student assessment

## Abstract

Both paper-based and computerized exams have a high level of cheating. It is, therefore, desirable to be able to detect cheating accurately. Keeping the academic integrity of student evaluations intact is one of the biggest issues in online education. There is a substantial possibility of academic dishonesty during final exams since teachers are not directly monitoring students. We suggest a novel method in this study for identifying possible exam-cheating incidents using Machine Learning (ML) approaches. The 7WiseUp behavior dataset compiles data from surveys, sensor data, and institutional records to improve student well-being and academic performance. It offers information on academic achievement, student attendance, and behavior in general. In order to build models for predicting academic accomplishment, identifying at-risk students, and detecting problematic behavior, the dataset is designed for use in research on student behavior and performance. Our model approach surpassed all prior three-reference efforts with an accuracy of 90% and used a long short-term memory (LSTM) technique with a dropout layer, dense layers, and an optimizer called Adam. Implementing a more intricate and optimized architecture and hyperparameters is credited with increased accuracy. In addition, the increased accuracy could have been caused by how we cleaned and prepared our data. More investigation and analysis are required to determine the precise elements that led to our model’s superior performance.

## 1. Introduction

Exam monitoring and its techniques are gaining more and more attention. Universities and academic institutions worldwide are vying for the newest technology to detect cheating in exam rooms and provide a rigorous atmosphere [1,2,3]. Invigilators are typically hired to oversee the whole test procedure to assure exam management and catch exam cheating.

The supervisor, the exam center director, or any other authorized individual immediately steps in in case of confirmed or attempted cheating [2]. The applicant might be removed immediately even if they utilized a cheat sheet, phone, or other objects. Supervisors will frequently write a report outlining the scenario if they are confident in the facts. We believe that additional staff is sufficient to oversee students remotely rather than automatically in order to supervise more students [3]. Although this is theoretically conceivable, it cannot be scaled for a course when more students are sitting their face-to-face or distance test [4,5,6]. Complex logistics would be necessary for this. Creating a real-time cheating detection system to replace human efforts is now feasible because of scientific and technical breakthroughs, notably with various deep learning algorithms [7,8]. The numerous movements made throughout the exam will be able to be detected by the automated system [9,10,11]. In addition, it will be able to identify numerous illegal items throughout the tests and record the timing of the sequence for a supervisory review at a later time. ML techniques using several kinds of neural networks are used in the current study.

The uniqueness of cheating behavior in higher education exams compared to other levels of tests can be attributed to several factors. Firstly, higher education exams are often more complex and challenging than tests in other levels. They require students to have a deeper understanding of the material, apply critical thinking skills, and the exams often have open-ended questions [12]. This complexity can increase the pressure on students to perform well, which may lead some to resort to cheating to obtain good grades. Additionally, the stakes are higher in higher education exams. These exams are often worth a larger proportion of the overall grade, and their outcomes can have significant implications for students’ academic and professional futures. This increased pressure to perform well can lead some students to engage in cheating behavior to achieve good grades, even if they have not adequately prepared for the exam [13]. Furthermore, in higher education, students are expected to have developed a strong sense of academic integrity and ethical conduct. However, some students may face significant challenges in maintaining these standards, particularly in high-stress situations such as final exams. For instance, students who feel overwhelmed or underprepared for an exam may be more likely to cheat. Another factor that contributes to the uniqueness of cheating behavior in higher education exams is the difficulty of detecting cheating. In higher education, courses often require students to have a deep understanding of the material and demonstrate critical thinking skills. As a result, it can be challenging for educators to distinguish between genuine student work and cheating [14]. Additionally, in online exams, where students are not directly monitored, there is a higher risk of academic dishonesty.

The deep learning algorithm employed in this study is called LSTM [15]. It is considered the most effective approach for dealing with object detection and recognition challenges and may be used to resolve data categorization concerns accurately [12,13,14,15]. It is a technique for deep learning designed to understand two-dimensional input, such as audio or images [16]. The motivation came from how humans process and grow their visual perception to recognize or differentiate an object in a digital image. To classify labeled data, it employs supervised learning techniques. As well as detecting, segmenting, and classifying pictures, it is commonly used to discriminate between objects or viewpoints [17]. This technique may also determine what people are doing [18]. Pooling, convolutional, and fully connected layers—the three variations of the layer—are stacked to construct it [19]. There are a variety of Convolution Neural Network (CNN) architectures, which are also employed in many research articles as deep-learning techniques and are regarded as such. As the convolution layer’s filter thickness matches the input’s thickness, the design varies from previous CNN systems. Both depth-wise and point-wise convolutions are recognized [20]. The bottleneck is where inputs and outputs between models occur, while the inner layers represent the model’s ability to accept inputs from lower-level ideas. Faster training and greater accuracy are possible via bypasses around bottlenecks [4].

The generalizability of these results is constrained despite the efficiency of the suggested approach. Through diligent work and study, it is unquestionably realistic and viable for a student to earn an extraordinarily high score [21]. Hence, a human expert must do more research before a final judgment is made in any situation flagged as a potential infringement. Notwithstanding its shortcomings, this research intends to add to this expanding field of study and provide insightful information on detecting final exam fraud.

Figure 1 displays the framework of the proposed model for detecting student cheating. Initially, all datasets go through data preparation and cleansing. Following that, the image pre-processing unit, feature extraction, and model selection activities are carried out. The model evaluation metrics are given to the classifier, which is then used to train the model. The optimization approach employs Support Vector Machine (SVM), LSTM, and Recurrent Neural Network (RNN) classifiers. The system model is then applied to identify whether a student is cheating or not.

The current study uses an internet protocol network detector and a behavior detection agent based on ML to solve the limits of online test cheating. The study was a case study, and its findings present ways to enhance the intelligent tutoring system. These are this study’s primary contributions:To identify online cheating using ML methods, we suggested a cheating intelligence agent based on the 7wiseup behavior dataset. We use the LSTM network with a densely linked idea, DenseLayer, and LSTM to construct behavior detectors. We selected cutting-edge ML methods for the online tests since they have developed quickly and have been extensively employed recently. They may offer helpful insights that contribute to the research field of an intelligent teaching system.Records were gathered from online examinations taken throughout mock, midterm, and final exam periods in highly uncontrolled settings. The database contained testing and training programs to analyze and evaluate performance.

The following is how the paper is structured: The second section is the literature review. Section 3 contains the dataset, followed by Section 4, which has the methodology part. Section 5 and Section 6 pertain to the results and discussion sections, respectively. Section 7 summarizes the concluding section.

## 2. Literature Review

Academic dishonesty is a challenging issue typically thought to happen when students use unethical writing practices, such as copying, plagiarism, pasting, glancing at other people’s work, and data falsification [2,3]. Due to faulty academic assessments and perhaps misleading student grades, academic dishonesty threatens the credibility of educational institutions [10,22]. Cheating on academic assignments is a significant ethical violation that jeopardizes academic integrity. Academic dishonesty has a significant negative influence on both the student’s ability to be trusted and the reputation of educational institutions. Educational institutions may ensure that their students are held accountable for their work by utilizing technology, such as digital essay scanning, turnitin.com, or software, to identify plagiarism [3,9,23]. Research in this area has advanced thanks to technological development.

The authors of the study [1] introduce a brand-new paradigm for the understanding and categorization of cheating video sequences. This sort of research assists in the early detection of academic dishonesty. The authors also present a brand-new dataset called “activities of student cheating in paper-based tests.” The dataset comprises suspicious behaviors that occurred in a testing setting. Eight different actors represented the five various types of cheating. Each pair of individuals engaged in five different types of dishonest behavior. They ran studies on action detection tasks at the frame level using five different kinds of well-known characteristics to gauge how well the suggested framework performed. The results of the framework trials were spectacular and significant.

In earlier research [12], the authors used the fact that students create their writings on electronic devices and used recorded keystrokes from assignments and exams to identify authors. By calculating the Euclidean distance between them, typing profile vectors from the students are compared. It is unrelated to what is written and functions in writing and programming duties. This method has the drawback of requiring additional software on the students’ devices to monitor their typing habits.

In order to identify cheating during an online test, the authors of [5] developed a system for assessing the head position and time delay. A high statistical correlation between cheating activity and a student’s head position change relative to a computer screen was also discussed. Therefore, we can instantly spot dubious student activity in online courses. Similarly, in [17], the authors suggested a novel technique for tracking a student’s anomalous conduct during an online test that uses a camera to establish the link between the examinee’s head and mouth. According to experiments, an irregular pattern of conduct in the online course may be easily identified using the suggested strategy.

In addition, the methods used by students to spot plagiarism in online tests were examined. To identify and discourage test cheating, the authors of [20] proposed an electronic exam monitoring system. The eye tribe tracker and the fingerprint reader were employed for continual authentication by the system. Due to this, the system used two factors to determine whether an examinee was cheating: the amount of time they were on-screen overall and how frequently they were off-screen. Keystroke dynamics’ importance in preserving security in online tests was discussed by [4]. Using statistical verification, ML, and logical comparison as its three stages, the suggested system employed authentication. An applicant’s typing style is immediately detected when he signs in for the first time, and a template is created for him. These templates are used as a reference to ensure the user is always authenticated when taking an online test. They are based on some characteristics, including dwell time (the time between pressing and releasing keys), flight time (the time between key releases and the next keypress), and the user’s typing speed for improved precision and responsiveness. The security risks related to online exams experienced in the past are discussed in [16] in their article published in volume seven. Complicity, which frequently entails the cooperation of a third party that helps the student by impersonating him or her online, was identified as a threat that was becoming more difficult to deal with. The probable mechanisms of security threats in online cheating were uncovered by a subsequent investigation conducted by the same authors [17,18]. Using dynamic profile questions from an online course, scientists evaluated the behavior of 31 individuals who took the test while being observed online. The findings revealed that students who cheated by impersonation exchanged most of the material through a mobile device. As a result, their reaction times were considerably different from those of non-cheaters [4].

The authors of [24] employed a different strategy involving hardware. The gear for the system comprises a camera, a wearable camera, and a microphone to keep track of the testing site’s visual and auditory environment [24]. Their research describes a multimedia analytics system that automatically gives out online tests. The system comprises six core parts that constantly evaluate the most significant behavioral cues: user verification, text detection, voice detection, active window detection, gaze estimation, and phone detection. In order to classify whether a test-taker is cheating at any point throughout the exam, we combined the continuous estimating components and added a temporal sliding window [25].

The authors of [26] used a case study to assess the incidence of possible e-cheating and offer preventative strategies that may be used. The internet protocol (IP) detector and the behavior detector are the two main components of the authors’ e-cheating intelligence agent, which they used as a technique for identifying online cheating behaviors. The intelligence agent keeps an eye on the students’ actions and is equipped to stop and identify any dishonest behavior. It may be connected with online learning tools to track student behavior and be used to assign randomized multiple-choice questions in a course test. This approach’s usefulness has been verified through testing on numerous datasets.

The references from the past about the subject of utilizing ML to identify student cheating are shown in Table 1.

Integrating computer vision using an ML technique is a crucial component of research breakthroughs, in addition to the hardware. By utilizing developments in ML, computer vision has become smore adept at processing pictures, identifying objects, and tracking, making research more precise and trustworthy. Cheating on an exam is usually thought of as an unusual event. Researcher identification is aided by peculiar postures or movements [26,27,28]. The application of computer vision efficiently enables this detection. Systems develop more intelligence through ML. Computer vision systems are now more capable of spotting suspicious actions because of this improved intelligence. ML processing technological developments also directly influence the results [29,30,31,32]. The use of this technique to identify suspicious behavior in both online and offline tests has been documented in many studies. CNN [33,34,35,36] is where most strategies were taken from.

**Table 1 sensors-23-04149-t001:** List of Past References, including methodology, dataset, techniques, and results.

Ref.	Dataset	Methodology	Results
[1]	Detail of the student cheating dataset: - Number of video sequences: 38 - Each class has an average number of images: 1660. - Testing images for each class: 510. - Unique subject recorded: Eight	Feature Extraction, ML, MSER Features, SURF, HOG, Robust Features.	SURF accuracy: 92% MSER Accuracy: 89% HOG Accuracy: 87%
[12]	Total Images of the Dataset: 7600.	3D CNN, Deep-Learning, Cheating in Exams, ML, Gesture Recognition Model, Object Detection.	- Lstm Model Accuracy: 0.77 - RNN Model Accuracy: 0.73 - 3DCNN Model Accuracy: 0.94.
[16]	- Online Participants from 5 different countries: 31 Participants. - Where Twenty-one students answered 379 questions.	Usability Online Examination,	- Students’ answers were 99.3% correct.
[24]	Students took part in the Exams: 104 students. - For Lab exams: six groups of up to twenty-two members were included.	Learning Management System, Learning Analytics. Python Tool	- Around twenty-three percent of students failed the test. - Whereas the previous result showed that forty-five percent of students failed.
[31]	7WiseUp dataset includes ninety-four students	Deep Neural Network (DNN), LSTM, DenseLSTM, RNN, Learning Management System (LMS) System	- RNN accuracy: 85%. - DenseLSTM accuracy: 96% - DNN accuracy: 67%. - LSTM accuracy: 93%.

The proposed approach overcomes these research gaps by utilizing a deep learning model that uses LSTM layers with dropout and dense layers to identify exam cheating among students. This approach is based on the use of ML technology and is more advanced than previous approaches that mainly relied on computer vision systems to detect cheating incidents.

The selection of LSTM as the technique for classifying cheating behavior of students was based on several reasons. Firstly, LSTMs are designed to handle sequential data, making them a natural choice for our time-series data consisting of sequential snapshots of student activity during the test. Secondly, LSTMs are known for their ability to capture long-term dependencies in sequential data, which is important in detecting cheating behavior that may involve multiple actions occurring over an extended period of time. Thirdly, LSTMs are capable of handling variable-length input sequences, which is necessary in a scenario where the number of actions a student takes during a test may vary. Fourthly, LSTMs are stateful models, which can be useful in detecting cheating behavior occurring over multiple input sequences. However, the LSTM technique was selected for its ability to handle sequential data, capture long-term dependencies, handle variable-length sequences, and maintain an internal state, making it a suitable choice for our problem of classifying cheating behavior of students. Additionally, our approach uses students’ grades in various exam portions as features in the dataset and labels them as “normal” or “cheating,” which improves anomaly identification methods. The detailed description of the proposed methodology is presented in the subsequent sections.

## 3. Dataset

### 3.1. Data Collection

A dataset that is openly accessible and contains information about student behavior in a university environment is called the 7WiseUp behavior dataset. The 7WiseUp initiative, which seeks to enhance student performance in the classroom by identifying and addressing issues that affect behavior, gathered the data.

The collection contains information from several sources, including surveys, sensor data, and institutional records. Information on student attendance, academic performance, and social conduct is included. The information may be used to create models for forecasting academic achievement, locating kids who are at risk, and spotting problematic conduct. It is intended for use in research on student behavior and performance. The dataset is made available under a Creative Commons license, which permits reuse and redistribution, as long as credit is given. However, the collection contains sensitive information about specific persons, so adhering to ethical standards and ensuring the data are handled responsibly is critical.

### 3.2. Data Description

The 7WiseUp behavior dataset is a publicly available dataset used for research in the detection of cheating behavior among students during online exams. The dataset consists of activity logs of 110 students who took an online exam. The logs record the actions taken by the students during the exam, such as scrolling, clicking, typing, etc. The dataset includes a total of 440 activity logs, with each log containing data from four different periods of the exam: the first quarter (Q1), the second quarter (Q2), the third quarter (Q3), and the fourth quarter (Q4). Each log also contains the students’ scores on the exam, as well as a label indicating whether the student was cheating or not during the exam. The cheating behavior in the dataset includes activities such as copying and pasting, using external resources, and collaborating with others during the exam. The 7WiseUp behavior dataset is valuable for researchers who want to develop and test algorithms for detecting cheating behavior in online exams using machine learning techniques.

The experimental assessment of our proposed technique uses four different synthetic datasets and one real-world dataset. The purpose of the synthetic datasets is to resemble real-world instances of test fraud. The dataset used in this study are

Dataset 1: It includes 10 instances of cheating and 100 students with average grades. Cheating incidents are regarded as blatant when there is a 35-point difference between the final test score and the average of the regular assessment results. The dataset has regular grades that increase during the semester. Every exam, including the final, has a 10-point scale, with the average score falling somewhere in the neighborhood of 80 percent. The cheating instances in Dataset 1 are rather simple to identify.Dataset 2: This dataset conceals cheating cases more effectively than Dataset 1, with just a 20-point difference between the average score before the final and the score on the final exam. A narrower gap between the final test and the prior results would make it more difficult to spot cases of cheating.Dataset 3: It has similarities with Dataset 2 but includes regular grades that climb for the semester. The red marks, however, unexpectedly increase by a large margin in the final exam. The average score before the final and the outcome of the final test differ very little in both circumstances.Dataset 4: This dataset simulates a straightforward final test where all the marks increase compared to the baseline assessments. On the final exam, the typical grades are simulated to increase by 10 points over the mean of the previous scores. The cheating occurrences are meant to increase test results on the final exam by 25 points compared to the preceding regular semester exams. Since all grades are improved for the final test, detecting cheating is more difficult.Real-world dataset: The real-world dataset used in the experiment includes 3 positive observations out of 52 total observations. Each observation includes the results of four quizzes, a midterm exam, and the final exam.

Human judgment has traditionally made the distinction between the first and second situations. However, we show that the recommended approach may automatically identify cheating instances even in these challenging conditions. The final dataset from our experiment illustrates the situation of a straightforward final test where all the marks increase compared to the baseline assessments. On the final exam, the typical grades are simulated to increase by 10 points over the mean of the previous scores. The cheating occurrences are meant to increase test results on the final exam by 25 points compared to the preceding regular semester exams. Since all grades are improved for the final test, detecting cheating is more difficult. In addition to the synthetic data, one real-world dataset is used in our experiments.

It would have been wonderful to have more real-world data, but it is not easy to collect for several reasons. A total of 3 of the 52 observations in our sample are positive. Each observation includes the results of four quizzes, a midterm exam, and the final exam. Figure 2 shows the high cheating dataset of Student 1 with normal grades. In this dataset, a normal score is considered to be within the range of 60–100.

The high cheating dataset of students that committed cheating is shown in Figure 3. According to the graph, 85% of students that take final examinations were found to cheat. At around 50%, Q4 had the lowest percentage of cheating pupils.

For a student with average grades, Figure 4 shows a graph with a smaller dataset of cheating. In this dataset, a normal score is considered to be within the range of 60–100. Around 58% of students in Q3 achieved normal grades. In contrast, pupils achieved about 50% in Q4. Nevertheless, pupils achieved 56% in their final exams.

A graph from the students who cheated in the less cheating dataset is shown in Figure 5. Another 60% of pupils were discovered to be cheating in Q4. At final exams, however, cheating was discovered in almost 85% of the pupils.

The graph for the dataset of students who were discovered to be cheating that involved less cheating and greater grades is shown in Figure 6. A total of 90% of the students who attempted to cheat on the final examinations were detected, on average. On the other hand, cheating is shown in Figure 7, which shows the graph of the decreasing cheating increasing grades of dataset 2, and the same is true: almost 85% of students were discovered cheating.

There are four distinct examples, each with two different courses where cheating was found in the student’s final grades. After viewing each student with normal and cheating grades, we notice that the abrupt surge in grades shows where the kid cheated. The dataset needed to train this model will be created in the next phase.

### 3.3. Data Preparation and Cleaning

Finding any null entries in the dataset and determining our characteristics will help with data preparation and cleaning. Ensuring the data are correct, comprehensive, and available for analysis is accomplished through data preparation and cleaning. The following are some typical procedures for cleaning and preparing data:Determine missing values: The following process was carried out on the dataset: Firstly, a check was made for any blank or missing values. Results lacking the necessary information may require greater precision. To locate any missing values, the isnull() and info() methods in pandas were utilized. Fortunately, no null entries were discovered in the dataset.Handling missing values: After locating missing values, there are several ways to deal with them. One way is to delete rows or columns with missing values or impute the missing values using the dataset’s mean, median, or mode. To impute missing values in pandas, the fill() method can be used. Fortunately, all entries in our dataset contain numerical characteristics and there are no missing entries.Check for duplicates: We have performed the following process on our dataset: Firstly, we checked for duplicates by ensuring that no rows or columns in the dataset are repeated. Results that contain duplicates may be skewed or erroneous. To find and eliminate duplicates, the pandas methods such as duplicated() and drop duplicates() is used. However, since these student ratings can be similar, we will modify the dataset later to remove any duplicates.Normalize the data: Normalizing the data will guarantee that all variables are scaled equally. The data were normalized to ensure that all variables are scaled equally. Normalizing the data is crucial, especially when using models that are sensitive to the data’s size. To normalize the data, we used the sci-kit-learn’s StandardScaler() method. For instance, exam scores are a collection of features in the dataset that are entirely numerical and were converted using the standard scalar option.Feature selection: We have performed the following process on our dataset: Firstly, feature selection was performed to choose the most relevant features for the investigation. This reduces the dimensionality of the dataset and increase the model’s accuracy. To choose the most relevant features, sci-kit-learn routines such as SelectKBest() or RFE() were used. Since the dataset is small, the four datasets were first concatenated and the features from Q1, Q2, midterm, Q3, Q4, and final were selected.

Any category variables should be converted to numerical variables. Many ML models use just numerical data. To convert categorical data to numerical variables, pandas provides methods such as LabelEncoder() and get dummies(). The “detection” column, which contains two classes—normal and cheating—is added to the label. These labels are first transformed into categorical data using a label encoder and then category instructions.

## 4. Proposed Methodology

Developing a deep learning model for detecting cheating in students involves several phases, which are outlined below, to ensure its success. To start, exploratory data analysis (EDA) is conducted to identify any irregularities or patterns in the dataset. Through this, information can be presented, statistical tests can be computed, and any missing or incorrect numbers can be detected. After running EDA, data cleansing and preparation must be performed to clean and prepare the dataset for analysis. This stage involves normalizing the data, handling missing values, and transforming categorical variables into numerical variables. Feature engineering is the next step; new characteristics are generated using current data that may be more useful or predictive in identifying cheating. Feature selection follows, where the most useful traits for spotting cheating are determined using statistical analysis or machine learning methods. The selected features are then used to choose a suitable machine learning method for identifying cheating. The choice of algorithm may be influenced by the size and complexity of the dataset, as well as the specific aims of the research. During model training, the chosen model is trained on a portion of the dataset, and its parameters are tweaked to maximize its performance. Following model training, model evaluation is done using a different validation set, and the model’s performance is assessed using F1 score, accuracy, recall, precision, and other performance indicators. Finally, hyperparameter tuning is conducted to improve the model’s performance on the validation set by changing its regularization, learning rate, or other model parameters to boost its performance.

The proposed methodology for the student cheating detection system is depicted in Figure 8. It comprises the retrieved characteristics from the 7WiseUp datasets. The method is separated into three levels, each with its own set of characteristics.

### 4.1. Model Architecture

The model we used to use the deep learning LSTM model to identify student cheating is listed below.

With the use of marks from several tests and quizzes, this sequential model architecture in Keras for binary classification aims to determine whether a student has cheated. Below is a thorough explanation of each layer:LSTM Layer 1: A layer called LSTM with two hidden units makes up the top layer. This layer delivers a series of output values for each input sequence in the shape of the training data it received as input. The output sequence, in this instance, is a set of hidden states that identify patterns in the sequence of grades, whereas the input sequence is a series of grades for a particular student.LSTM Layer 2: A LSTM layer with 1 hidden unit also makes up the second layer. The output sequence from the first LSTM layer is passed on to this layer, producing a final output value for the complete series. In order to create a single output value that reflects the whole range of grades, the information from the previous layer is combined in this layer.Dropout Layer: A dropout layer with a dropout rate of 0.8 makes up the third layer. To avoid overfitting, 80% of the connections between the second LSTM layer and the following layer are randomly dropped out in this layer.Dense Layer 1: The first dense layer is the fourth layer, with two hidden units and a ReLU activation function. In this layer, the input is transformed linearly, and the output is activated using a rectified linear unit (ReLU).Dropout Layer: At a dropout rate of 0.2, the fifth layer is also a dropout layer. This layer randomly removes 20% of the connections between the previous dense layer and the final output layer to prevent overfitting.Dense Layer 2: With one output unit and a sigmoid activation function, the second dense layer is the sixth layer. The input is transformed linearly inside this layer, while the output is activated sigmoidally. Based on a student’s grades, the sigmoid function generates a probability value between 0 and 1, indicating how likely cheating is.

In order to capture the sequential patterns in the grades data, our model architecture consists of two LSTM layers, two dense layers with dropout to minimize overfitting, and a final output layer with sigmoid activation to predict the likelihood of cheating. The model is tested using a different validation set after being trained on the training data to improve performance. As shown in the below Figure 9. 

#### 4.1.1. LSTM

The RNN architecture known as LSTM, or long short-term memory, is used to model sequential data. To better capture long-term relationships in the sequence, LSTM networks, unlike conventional RNNs, utilize a memory cell to retain data about the prior inputs and their dependencies. There are gates in the memory cell that manage the flow of information. These gates, which include an input gate, an output gate, and a forget gate, govern how information is added to, extracted from, and deleted from the cell.

We have implemented an LSTM network for tasks that involve sequential data such as time series analysis and voice recognition. LSTM networks are particularly useful for input sequences with long-term dependencies as they can selectively retain and discard information over time; this is unlike classic RNNs that suffer from the “vanishing gradient problem” when the input sequence is long. We have utilized two LSTM layers in our model. The first layer consists of two units and is initialized with the LSTM layer type. The input shape parameter is defined to match the shape of a single sample from the training data. Additionally, we have set the return sequences option to True, which allows the layer to produce a sequence of hidden state values instead of a single output value. The second layer in our model is also an LSTM layer but only contains one unit. By default, the return sequences option for this layer is set to False, which means that it outputs a single value for each sequence. By implementing these LSTM layers, we can effectively handle tasks that involve sequential data and long-term dependencies.

#### 4.1.2. Dropout

We have implemented the dropout regularization approach in the proposed model to address the issue of overfitting. Dropout is a technique that randomly removes a portion of the input units during training to prevent the model from learning to match the training data too closely, which can lead to poor generalization performance on new, unseen data.

We applied dropout to a layer’s input or hidden units with a predetermined dropout rate, typically between 0.2 and 0.5. During training, a unit is likely to be dropped out in any iteration if the dropout rate is high enough. The remaining units’ weights are scaled by the inverse of the dropout rate to compensate for the discarded units during training. During testing, the complete network is used without dropout.

The dropout technique allows the network to acquire more reliable and generalizable input representations by randomly removing units during training. This technique helps to prevent the network from relying too heavily on certain input properties, thereby increasing the model’s ability to generalize to new data.

To implement the dropout technique in our DL model, we used a dropout layer with a dropout rate of 0.8. During training, this dropout layer randomly drops out a portion of the input units, introducing noise to the network, and reducing its dependence on specific input qualities, thus helping to prevent overfitting. With a dropout rate of 0.8, training would lose 80% of the input units, further aiding in generalization.

#### 4.1.3. Dense Layers

We have utilized dense layers which are also known as “completely connected layers.” Each neuron in a dense layer receives input from every neuron in the previous layer through its output. A linear operation is first applied to the input by the dense layer, followed by a nonlinear activation function. The output of the dense layer is a series of weighted sums of the inputs, where each sum corresponds to a distinct neuron in the layer. We have used backpropagation to train the weights and biases associated with each neuron in the dense layer. Dense layers are commonly used in neural networks for applications such as image classification, natural language processing, and speech recognition. They are often combined with other layers such as convolutional or recurrent layers to build more complex network topologies.

We have also used a fully connected (dense) layer with two units and a ReLU activation function to incorporate non-linearity into the network and enhance its capacity to model complicated relationships in the data. Additionally, we have added another dropout layer with a lower dropout rate of 0.2 compared to the third layer. Finally, the network’s top layer is a fully linked (dense) layer with a sigmoid activation function. The sigmoid activation function translates the output into a probability value between 0 and 1, which represents the likelihood that the input belongs to the positive class.

### 4.2. Model Hyperparameters

In the model.compile() method, the following hyperparameters are used for the specified architecture:Loss: In binary classification issues, the loss function known as binary cross-entropy is employed. It calculates the discrepancy between the actual labels for each sample in the training set and the anticipated probabilities for each. The loss function assesses the model’s performance during training.Optimizer: Adam is a well-known technique for optimizing the adaptive learning rate in DL. It boosts the speed and stability of gradient descent by combining the benefits of AdaGrad and RMSProp, two additional optimization techniques. The neural network’s weights are adjusted during training by the optimizer’s specifications.Metrics: A performance indicator called accuracy counts how many instances in the validation set out of all the examples were properly categorized. It tests how accurately the model can discriminate between honest and dishonest groups through cheating.

The decision made about the optimizer, loss function, and performance metric may significantly impact the model’s performance. Binary cross-entropy is a logical option for a binary classification task. The Adam optimizer is well recognized as suitable for many deep-learning models. The accuracy metric gives a straightforward and understandable indication of how well the model performs. However, alternative metrics such as precision, recall, or F1 score could be more suitable depending on the application’s objectives. The hyperparameters may be adjusted by performing several tests with various settings and choosing the one that produces the best results.

## 5. Results

The process of evaluating a trained ML model on a dataset different from the data used for training is known as model evaluation. Python is a popular programming language that is widely used in data science and ML. In our research, we used Python version 3.8.5 for our implementation. To aid our implementation, we made use of several popular libraries such as TensorFlow, Keras, Pandas, and Numpy. These libraries helped us to carry out various tasks such as data preprocessing, building and training our deep learning model, and evaluating our results. We specifically used TensorFlow version 2.4.0, which is a popular open-source platform for machine learning and deep learning. We also used Keras version 2.4.3, which is a high-level API built on top of TensorFlow that simplifies the process of building and training deep learning models. Pandas version 1.1.3 was also used to manipulate and analyze our dataset. Finally, we made use of Numpy version 1.19.2 to perform numerical computations on our dataset.

We have followed several procedures to evaluate the performance of the LSTM model. Firstly, we split the dataset into training and testing sets using the train–test split function from the sci-kit-learn library. The testing set was used to evaluate the model’s performance after it was trained on the training set. Next, we applied the LSTM model to the training set using the fit() technique to train the algorithm to predict each student’s likelihood of cheating based on their performance on the Q1, Q2, midterm, Q3, Q4, and final exams. After training, we evaluated the model on the testing set to predict the probability of cheating for each student. We calculated evaluation metrics such as accuracy, precision, recall, F1-score, Receiver Operating Characteristic Area Under the Curve (ROC-AUC), etc., using the predicted probabilities and actual labels. The evaluation metrics indicated how well the model distinguished between normal and detected cheating in the two groups. To enhance the LSTM model’s performance, we adjusted hyperparameters such as the number of LSTM layers, neurons in each layer, dropout rate, learning rate, and batch size. The hyperparameters were chosen based on the model’s performance on the validation set. Finally, we visualized the results using charts such as the ROC curve, confusion matrix, and precision–recall curve. These plots helped to identify areas that needed improvement and provided insights into the model’s strengths and weaknesses. By following these procedures, we effectively assessed the performance of the LSTM model.

By carrying out the processes listed above, the LSTM model’s performance may be assessed and enhanced for better outcomes on the dataset for detecting student cheating.

After 50 iterations, the model achieved 90% training and 92% validation accuracy.

The performance of a DL model during training may be assessed using measures such as training and validation accuracy. Whereas validation accuracy refers to the model’s performance on a different validation set that is not used for training, training accuracy refers to the model’s performance on the training set. Figure 10 demonstrates the model’s training accuracy, with a blue graph for training and an orange graph for validation.

The model should be neither overfitting nor underfitting the data if its training and validation accuracies are good and near to one another. It may be a sign of overfitting, where the model has learned the training data too well and cannot generalize to new data, if the training accuracy is significantly greater than the validation accuracy. On the other hand, if the accuracy of both training and validation is poor, it may be a sign that the model is underfitting, which means it cannot capture the patterns in the data.

We can experiment with alternative model topologies, hyperparameters, and optimization strategies to increase validation accuracy. To reduce overfitting and increase the generalizability of the model, we may utilize strategies such as regularization, early halting, and data augmentation. Moreover, we can expand the training dataset or utilize strategies such as transfer learning to use previously learned models. Our training accuracy is over 90%, while our validation accuracy is around 92%.

A DL model’s performance during training may be assessed using metrics such as training and validation loss. The difference between training loss and validation loss is that the former refers to the model’s average loss on the training set. In contrast, the latter refers to the model’s average loss on a unique validation set not used for training. The model training loss is shown in Figure 11 as a blue and an orange graph, respectively, where the blue graph represents the training loss.

We want training and validation loss to be minimal and somewhat near to one another, similar to how training and validation accuracy should be. High validation loss may signify overfitting or the inability of the model to generalize to new data, whereas high training loss may suggest that the model is unable to adequately fit the training set of data.

Alternate model topologies, hyperparameters, optimization methods, regularization approaches, early halting, and data augmentation are some techniques to reduce the validation loss. These techniques were previously described as ways to reduce validation accuracy. To improve the model’s generalization, we could also expand the size of the training dataset or employ methods such as transfer learning. Furthermore, we can keep track of the loss experienced during training and change the learning rate or batch size as necessary.

Due to the short dataset size and use of a deep learning model for training, our dataset has little overfitting. The accuracy would be lower but there would be no overfitting in the model if we had used classifiers such as SVM.

### 5.1. Model Evaluation Metrics

Metrics for measuring the effectiveness of a DL model on a specific task are known as model evaluation metrics. For binary classification tasks, the following assessment criteria are frequently used:Recall: The fraction of genuine positives in the test set out of all real positive samples.Accuracy: The model’s percentage of correct predictions on the test set.F1 score: Precision and recall’s harmonic mean is the F1 score.Precision: The fraction of true positives (positive samples accurately detected) out of all positive predictions produced by the model.ROC curve and AUC: The ROC curve represents the relationship between the true positive rate (recall) and the false positive rate (1-specificity) at various categorization thresholds. The ROC curve’s AUC measures how well the model can differentiate between positive and negative data.

The proposed model for the test data’s ROC curve is shown in Figure 12, where the ROC curve tracks the same path as the random guess graph. The false positive rate (FPR) and true positive rate (TPR) are shown in the graph.

### 5.2. Confusion Matrix

The confusion matrix summarizes the number of true positives, true negatives, false positives, and false negatives in the test set. The confusion matrix of the suggested test data model is displayed in Figure 13.

Many Python libraries, including scikit-learn and TensorFlow, may be used to calculate these metrics. It is critical to select the right assessment metrics depending on the particular job at hand and the dataset’s class imbalance.

The accuracy and loss performance are shown in Table 2, where training accuracy has a performance value of about 0.90, validation accuracy of about 0.92, training loss of about 0.39, and validation loss of about 0.35. Metrics for assessing the proposed model are shown in Table 3. The average accuracy in this instance is 0.91.

### 5.3. Model Comparison

We examined our model with three more reference works in our investigation of the student cheating detection dataset. With an accuracy of 84.52% on the 7WiseUp behavior dataset, the initial reference study employed a CNN technique. With an accuracy of 81% on the same dataset, the second reference study employed an LSTM technique. In the third reference study, the 7WiseUp behavior dataset was employed using an RNN method that had an accuracy of 86%.

In our examination of the student cheating detection dataset, we compared our model to three existing reference works. With an accuracy of 84.52% on the 7WiseUp behavior dataset, the initial reference study employed a CNN technique. The same dataset was utilized in the second reference work’s LSTM method, which had an accuracy rate of 81%. For the 7WiseUp behavior dataset, the third reference study employed an RNN method with an accuracy of 86%. The comparison is as shown in Table 4.

Our model design, which utilized an LSTM strategy with a dropout layer, thick layers, and an Adam optimizer, produced results that were 90% more accurate than those of all three-reference works. We credit our usage of more complex and optimized architecture and hyperparameters for greater accuracy. The greater accuracy is likely attributable to the data processing and cleaning we performed. More investigation and analysis are needed to pinpoint precisely what contributed to our model’s enhanced performance.

## 6. Discussion

Keeping the academic integrity of student evaluations intact is one of the biggest issues in online education. The absence of direct teacher monitoring significantly increases academic dishonesty during final exams. The 7WiseUp behavior dataset, which anybody can access, is used in this project to offer details on student conduct in a university environment. The 7WiseUp program gathered the data, which aim to improve student success by identifying and resolving issues that impact behavior.

A variety of sources, including surveys, sensor data, and institutional records, are included in the collection. It includes information on student attendance, academic progress, and social behavior. The information may be used to build models for predicting academic success, locating at-risk students, and spotting problematic behavior. It is designed for use in research on student behavior and performance [38,39,40,41,42,43,44,45,46,47,48,49].

The term’s dataset, final exam, suggested technique, final test, and average score all use four different synthetic datasets and one real-world dataset for the experimental evaluation of a suggested strategy. The final test score and the average of the regular assessment scores differ by 35 points, making datasets 1 and 2 equal. The last 20 regular grades improve during the semester; the final test result is within a 10-point range for around 80% of the average marks. A new label column with two classes—normal and cheating—has been added, improving anomaly identification methods. Our experiment’s final dataset demonstrates that cheating incidents can be automatically identified even in challenging situations. The cheating occurrences are designed to raise exam scores by 25 points and mimic a jump in average grades of 10 points over the mean of the preceding grades. Together with the synthetic data, we also employ one real-world dataset. For each observation, we include 1 of the 52 observations in our collection.

A CNN technique (84.52%), an LSTM approach (84.52%), and an RNN approach (86%), along with three additional reference works, were compared to the student cheating detection dataset. The first reference work used a CNN approach with 84.52% accuracy, the second used an LSTM approach with an error rate of 81%, and the third used one with an objective accuracy of 86%. All three-reference works were tested on the 7WiseUp behavior dataset. In contrast, the accuracy of our model architecture, which used an LSTM approach with a dropout layer, thick layers, and an optimizer called Adam, was 91%, which was higher than the accuracy of all three reference studies. Implementing more intricate and sophisticated architectures and hyperparameters is responsible for the increased accuracy. Additionally, our data preparation and cleaning process may be responsible for improved accuracy. More investigation and analysis are required to identify the specific factors that were responsible for our model’s superior performance.

One limitation of this research is that it relied on a single dataset, the 7WiseUp behavior dataset, which may not be representative of all online education environments. Furthermore, the dataset was not specifically designed for cheating detection, which may limit the accuracy of the models developed in this research. Additionally, the synthetic datasets used in the experiments may not fully capture the complexity of real-world cheating incidents. Further research could benefit from using multiple datasets, including those specifically designed for cheating detection, to ensure the generalizability of the findings. Another limitation is that the specific factors responsible for the superior performance of the model are not identified, highlighting the need for further analysis and investigation.

In addition, another limitation is low performance metric values for imbalanced data; the proposed model also has low values of recall and F1 score. Recall, also known as sensitivity or true positive rate, measures the proportion of actual positives that are correctly identified by the model. F1 score is a combination of precision and recall, and it takes into account both false positives and false negatives. These metrics are particularly important for imbalanced datasets, where the proportion of positive cases is much lower than that of negative cases. In such cases, a model that simply predicts all cases as negative may achieve high accuracy but perform poorly in terms of identifying positive cases. Therefore, low values of recall and F1 score indicate that the proposed model is not effective at identifying positive cases, which is a significant limitation for its applicability in real-world scenarios.

## 7. Conclusions

The rise of online education has presented many benefits for students and educational institutions, but it has also brought forth numerous challenges, including academic dishonesty in the form of cheating, during online assessments. To address this issue, educational institutions must implement better detection techniques to ensure academic integrity. This research uses ML technology to investigate the problem of online cheating and provides practical solutions for monitoring and eliminating such incidents. The goal of this research was to create a deep learning model using LSTM layers with dropout and dense layers to identify exam cheating among students. We used the students’ grades in various exam portions as features in our dataset and labeled them as “normal” or “cheating.” Despite having a smaller dataset than previous research, our model architecture resulted in a 90% training and 92% validation accuracy, outperforming models that used CNN and RNN layers. Our approach accurately and successfully identified student exam cheating, showcasing the potential of deep learning approaches in identifying academic dishonesty. By utilizing such models, educational institutions can create more efficient strategies for guaranteeing academic integrity. Ultimately, this research emphasizes the importance of using advanced technologies in addressing contemporary challenges in online education.

Future research should focus on further refining and optimizing deep learning models for detecting academic dishonesty in online assessments. This can include exploring the use of other machine learning algorithms and techniques, such as ensemble learning and transfer learning, to improve model performance and accuracy. Additionally, research can investigate fthe feasibility of implementing real-time monitoring systems that can detect and prevent cheating during online exams.

## Figures and Tables

**Figure 1 sensors-23-04149-f001:**
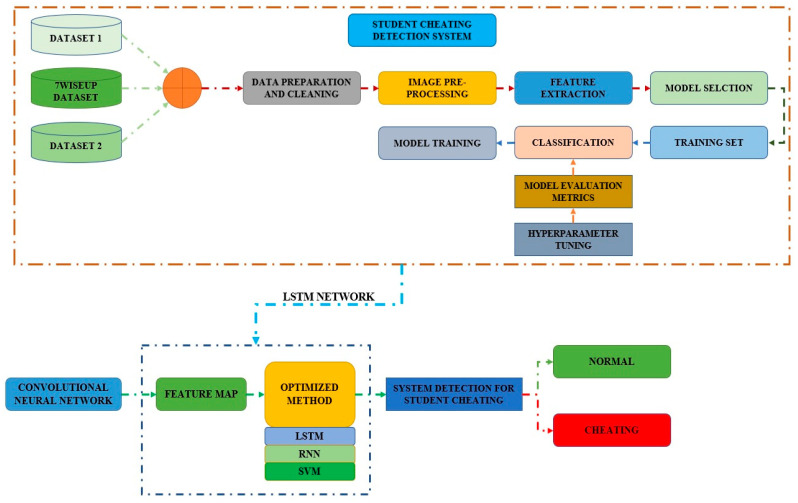
Framework of the proposed model diagram related to student cheating detection system.

**Figure 2 sensors-23-04149-f002:**
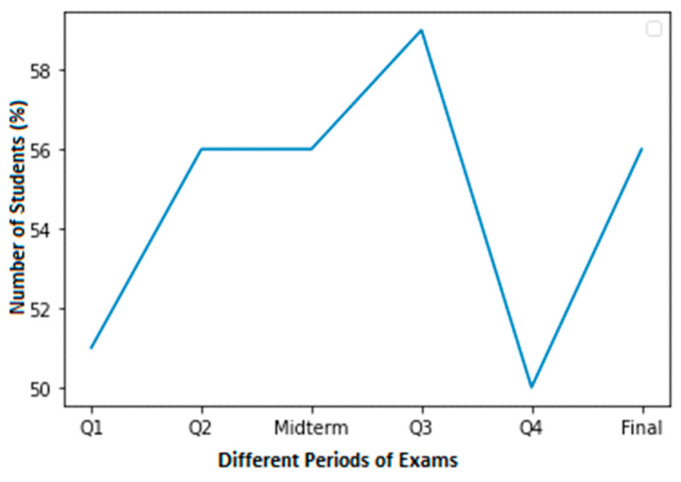
A high cheating dataset of student 1 with normal grades.

**Figure 3 sensors-23-04149-f003:**
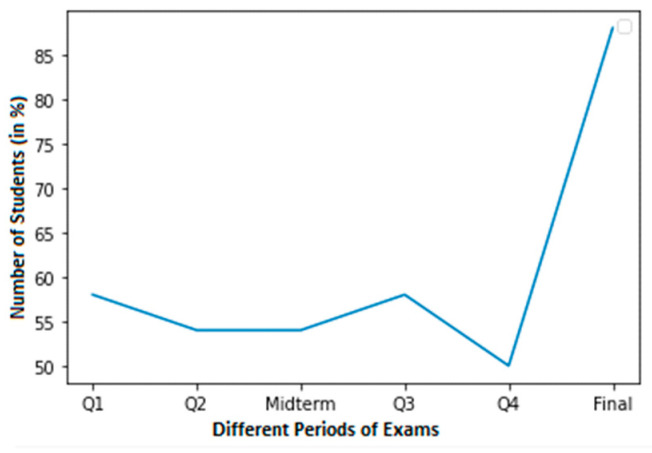
A high cheating dataset of students with cheating found.

**Figure 4 sensors-23-04149-f004:**
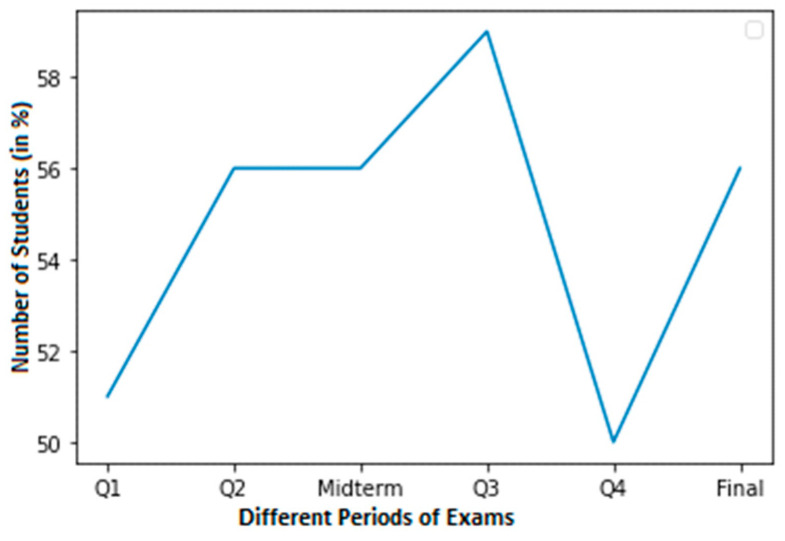
Less cheating dataset of a student with normal grades.

**Figure 5 sensors-23-04149-f005:**
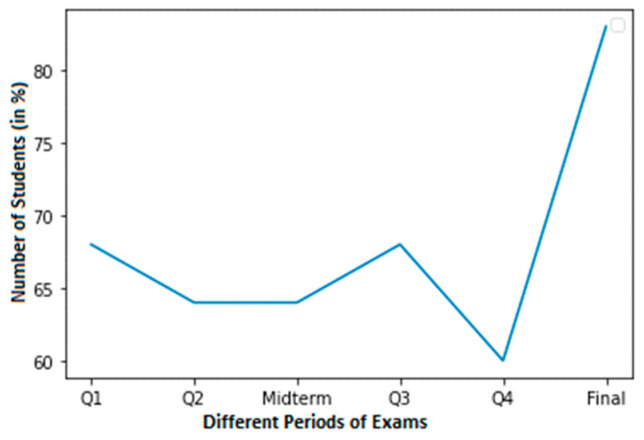
Less cheating dataset of students with cheating found.

**Figure 6 sensors-23-04149-f006:**
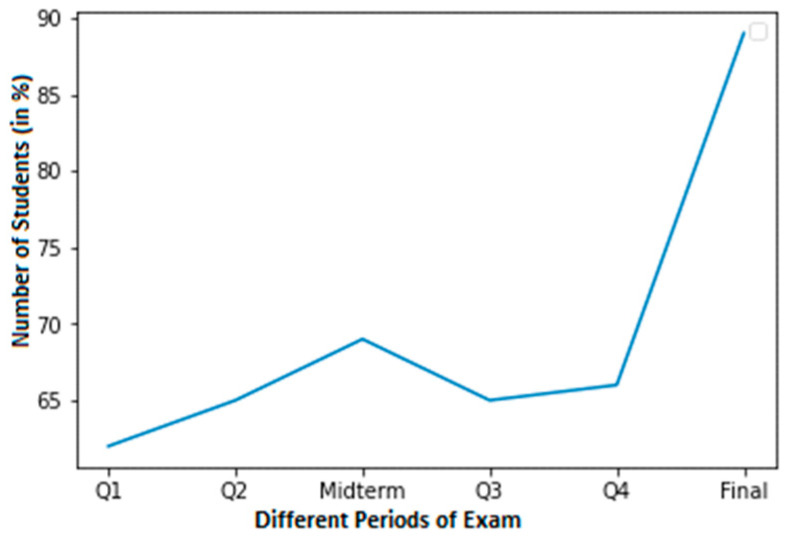
Less cheating increasing grades dataset of students with cheating found.

**Figure 7 sensors-23-04149-f007:**
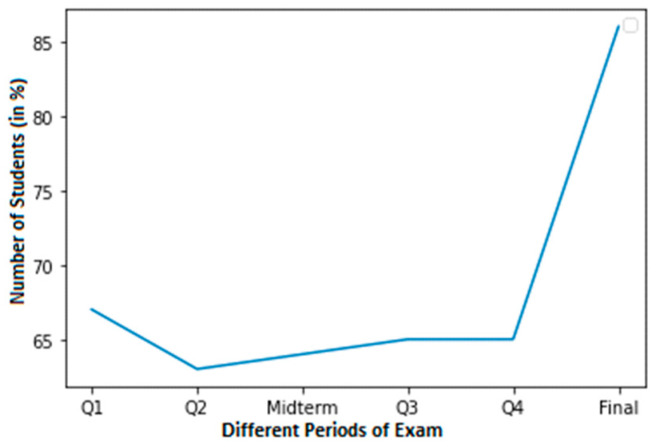
Less cheating increasing grades dataset 2 with cheating found.

**Figure 8 sensors-23-04149-f008:**
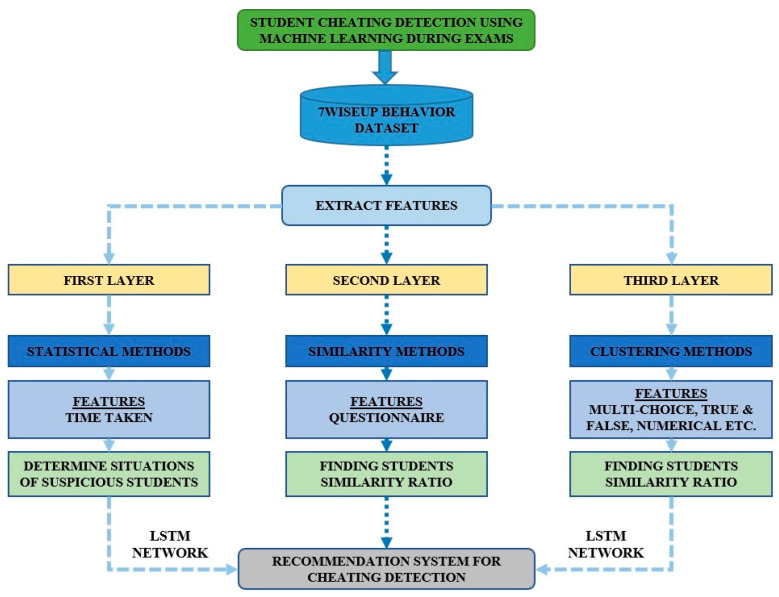
Proposed methodology for student cheating detection system.

**Figure 9 sensors-23-04149-f009:**
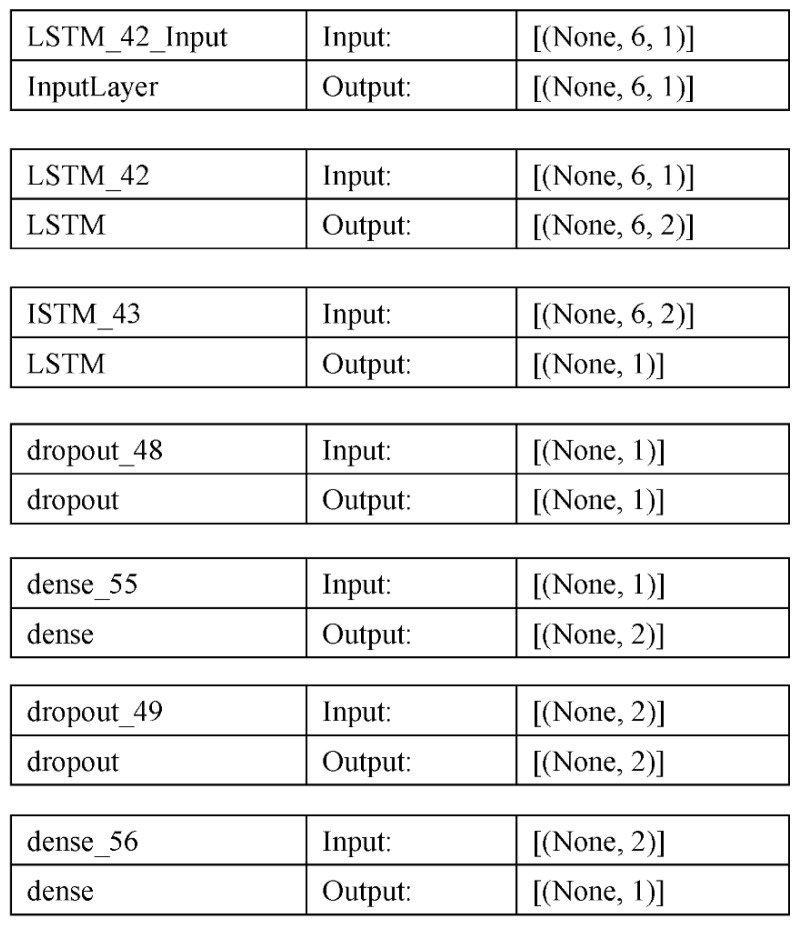
The model architecture of the proposed model.

**Figure 10 sensors-23-04149-f010:**
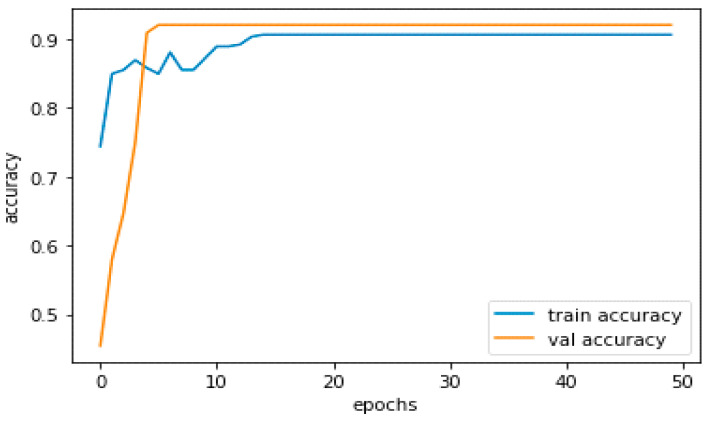
Model training accuracy.

**Figure 11 sensors-23-04149-f011:**
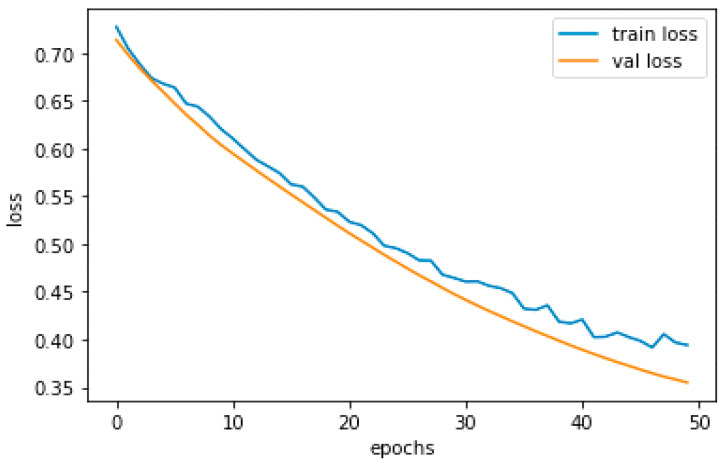
Model training loss.

**Figure 12 sensors-23-04149-f012:**
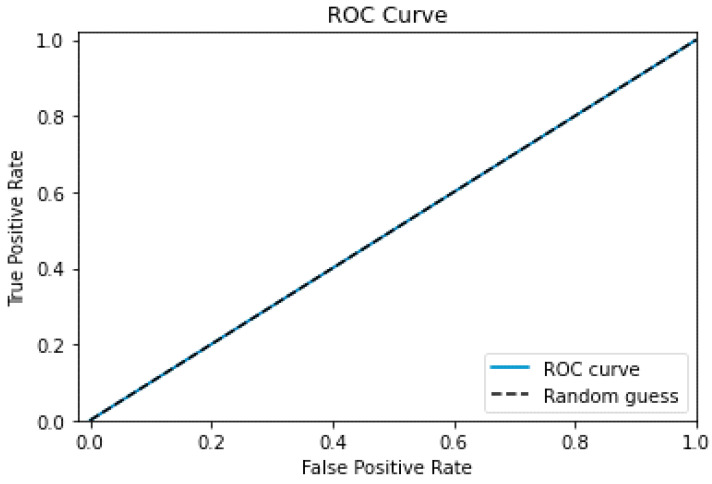
ROC curve of the proposed model for test data.

**Figure 13 sensors-23-04149-f013:**
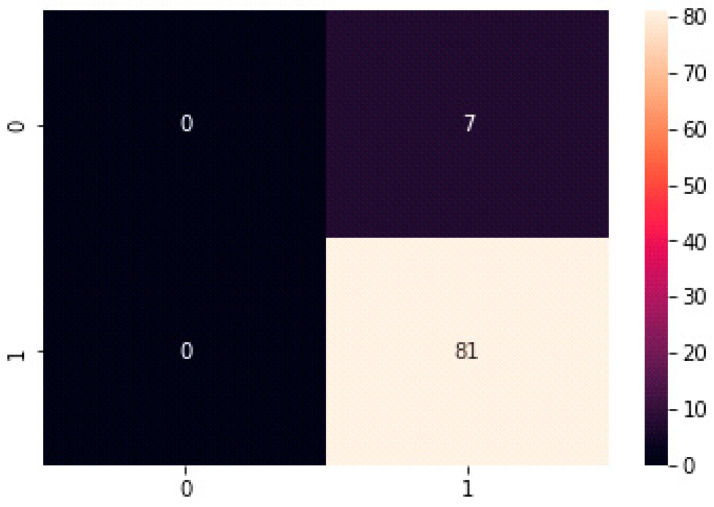
Confusion matrix of the proposed model of test data.

**Table 2 sensors-23-04149-t002:** Accuracy and loss performance.

Evaluation Metric	Performance Value
Training accuracy	0.90
Validation accuracy	0.92
Training loss	0.39
Validation loss	0.35

**Table 3 sensors-23-04149-t003:** Evaluation metrics of the proposed model.

Evaluation Metric	Performance Value
Mean accuracy	0.91
accuracy	0.92
precision	0.62
recall	0.52
F1 score	0.78

**Table 4 sensors-23-04149-t004:** Proposed model comparison with related work.

Ref	Approach	Accuracy	Dataset
[4]	CNN	86.07%	7WiseUp behavior dataset
[37]	LSTM	81%	7WiseUp behavior dataset
[31]	RNN	86%	7WiseUp behavior dataset
Our approach	LSTM	90%	7WiseUp behavior dataset

## Data Availability

The dataset used in this study is publicly available at http://7wiseup.com/research/e-cheating/data/ accessed on 20 March 2021.

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
