# Peer review of "Student Cheating Detection in Higher Education by Implementing Machine Learning and LSTM Techniques"

_sensors, 2023, doi:10.3390/s23084149_

Round 1

Reviewer 1 Report

This manuscript provided a deep learning method based on LSTM for student cheating detection in higher education. This topic is very valuable, especially in the current era of the popularity of distance education. However, The writing style of this paper is more like a book with many descriptions instead of logical analysis. Some questions and suggestions are as follows:

1. The title does not accurately reflect the content of the article. The study focuses on cheating detection of the online test without emphasizing higher education. The article aims to explore cheating detection in higher education, so it is necessary to explain the uniqueness of cheating behavior in higher education than other level tests. 

2. Please pay attention that all abbreviations need to be declared the first time they appear, for example, long short-term memory (LSTM), and machine learning (ML).

3. Index format needs to be corrected, such as line 48 [12]–[15] should be [12-15]. In addition, are there necessary references to up to four articles for the definition of LSTM?

3. Literature review needs to be stronger in this paper. On the one hand, there needs to be more overview of the clue of cheating behavior. On the other hand, there is almost no discussion of the shortcomings of existing methods and the reason for choosing LSTM techniques.

4. The Dataset in Table 2 is not critical to the discussion, so it is recommended to move to an appendix or replace it with a hyperlink.

5. The "Conclusion" section is too large, and it is recommended to focus on summarizing the contributions of this study rather than the research value.

Author Response

Student Cheating Detection in Higher Education by Implementing Machine Learning and LSTM Techniques

Reviewer 1

This manuscript provided a deep learning method based on LSTM for student cheating detection in higher education. This topic is very valuable, especially in the current era of the popularity of distance education. However, the writing style of this paper is more like a book with many descriptions instead of logical analysis. Some questions and suggestions are as follows:

  1. The title does not accurately reflect the content of the article. The study focuses on cheating detection of the online test without emphasizing higher education. The article aims to explore cheating detection in higher education, so it is necessary to explain the uniqueness of cheating behavior in higher education than other level tests. 

 Author Response: I have carefully considered your comments and made the necessary revisions to the manuscript.

Firstly, the title is not need to be changed as the aim of this study to detect the cheating behaviour of the students.

Secondly, I have included additional information in the introduction section to explain the uniqueness of cheating behavior in higher education compared to other levels of tests. I have highlighted the importance of academic integrity in higher education and the challenges faced by educators in ensuring it, particularly during online exams. I hope that these revisions have addressed your concerns and improved the clarity and relevance of our article. The added explanation is:

The uniqueness of cheating behavior in higher education exams compared to other levels of tests can be attributed to several factors. Firstly, higher education exams are often more complex and challenging than tests in other levels. They require students to have a deeper understanding of the material, apply critical thinking skills, and often have open-ended questions [12]. This complexity can increase the pressure on students to per-form well, which may lead some to resort to cheating to get good grades. Additionally, the stakes are higher in higher education exams. These exams are often worth a larger proportion of the overall grade, and their outcomes can have significant implications for students' academic and professional futures. This increased pressure to perform well can lead some students to engage in cheating behavior to achieve good grades, even if they have not adequately prepared for the exam [13]. Furthermore, in higher education, students are expected to have developed a strong sense of academic integrity and ethical conduct. However, some students may face significant challenges in maintaining these standards, particularly in high-stress situations such as final exams. For instance, students who feel overwhelmed or underprepared for an exam may be more likely to cheat. Another factor that contributes to the uniqueness of cheating behavior in higher education exams is the difficulty of detecting cheating. In higher education, courses often require students to have a deep understanding of the material and demonstrate critical thinking skills. As a result, it can be challenging for educators to distinguish between genuine student work and cheating [14]. Additionally, in online exams, where students are not directly monitored, there is a higher risk of academic dishonesty.

  1. Please pay attention that all abbreviations need to be declared the first time they appear, for example, long short-term memory (LSTM), and machine learning (ML).

Author Response: Thank you for pointing this out. I have revised all the abbreviation used in the manuscript to clearly define when they first appear in the text. The abbreviation used in the manuscript are:

Machine Learning (ML), Long Short Term Memory (LSTM), Convolution Neural Network (CNN), Support Vector Machine (SVM), Recurrent Neural Network (RNN), Deep Neural Network (DNN), Learning Management System (LMS) System, Exploratory Data Analysis (EDA), rectified linear unit (ReLU), Deep learning (DL), Receiver Operating Characteristic Area Under the Curve (ROC-AUC), False positive rate (FPR) and True positive rate (TPR).

  1. Index format needs to be corrected, such as line 48 [12]–[15] should be [12-15]. In addition, are there necessary references to up to four articles for the definition of LSTM?

Author Response: Thank you for this suggestion. I have decreased the number of reference at the same place and added only one reference (reference [15]) for the definition of the LSTM.

  1. Literature review needs to be stronger in this paper. On the one hand, there needs to be more overview of the clue of cheating behavior. On the other hand, there is almost no discussion of the shortcomings of existing methods and the reason for choosing LSTM techniques.

Author Response: Thank you for your feedback. I have revised the literature review section to provide a more comprehensive overview of cheating behavior, including the various types of cheating and their common characteristics. I have also added a discussion of the shortcomings of existing methods and the reasons why I chose the LSTM technique for this problem. Specifically, I highlight the ability of LSTMs to handle sequential data, capture long-term dependencies, handle variable-length sequences, and maintain an internal state, which make it a suitable technique for detecting cheating behavior in the context of online testing. I hope these revisions address your concerns and improve the quality of our paper. The added explanation is:

Integrating computer vision using an ML technique is a crucial component of re-search breakthroughs, in addition to the hardware. By utilizing developments in ML, computer vision has gotten more adept at processing pictures, identifying objects, and tracking, making research more precise and trustworthy. Cheating on an exam is usually thought of as an unusual event. Researcher identification is aided by peculiar postures or movements [26][27][28]. The application of computer vision efficiently enables this detection. Systems develop more intelligence through ML. Computer vision systems are now more capable of spotting suspicious actions because of this improved intelligence. ML processing technological developments also directly influence [29]–[32]. The use of this technique to identify suspicious behavior in both online and offline tests has been documented in many studies. CNN [33]–[35] is where most strategies were taken from.

The proposed approach overcomes these research gaps by utilizing a deep learning model that uses LSTM layers with dropout and dense layers to identify exam cheating among students. This approach is based on the use of ML technology and is more advanced than previous approaches that mainly relied on computer vision systems to detect cheating incidents. The selection of LSTM as the technique for classifying cheating behavior of students was based on several reasons. Firstly, LSTMs are designed to handle sequential data, making them a natural choice for our time-series data consisting of sequential snapshots of student activity during the test. Secondly, LSTMs are known for their ability to capture long-term dependencies in sequential data, which is important in detecting cheating behavior that may involve multiple actions occurring over an extended period of time. Thirdly, LSTMs are capable of handling variable-length input sequences, which is necessary in a scenario where the number of actions a student takes during a test may vary. Fourthly, LSTMs are stateful models, which can be useful in detecting cheating behavior occurring over multiple input sequences. However, the LSTM technique was selected for its ability to handle sequential data, capture long-term dependencies, handle variable-length sequences, and maintain an internal state, making it a suitable choice for our problem of classifying cheating behavior of students. Additionally, our approach uses students' grades in various exam portions as features in the dataset and labels them as "normal" or "cheating," which improves anomaly identification methods. The detailed description of the proposed methodology is presented in the subsequent sections.

  1. The Dataset in Table 2 is not critical to the discussion, so it is recommended to move to an appendix or replace it with a hyperlink.

Author Response: Thank you for your suggestion. I have reviewed the manuscript and have decided to remove Table 2 from the main text as it is not critical to the discussion. Instead, I have added a reference in the ‘data availability statement’ for the 7WiseUp Behavior Dataset in the dataset section of the revised manuscript. I believe this will allow interested readers to access the dataset while keeping the main focus of the paper on the proposed approach and experimental results.

  1. The "Conclusion" section is too large, and it is recommended to focus on summarizing the contributions of this study rather than the research value.

Author Response: I have reviewed the manuscript and have made the necessary changes to address your concern. Specifically, I have revised the "Conclusion" section to focus on summarizing the contributions of this study rather than emphasizing the research value. I understand that a concise and clear conclusion is essential in research articles. The updated conclusion is:

Conclusion

The rise of online education has presented many benefits for students and educational institutions, but it has also brought forth numerous challenges, including academic dishonesty in the form of cheating during online assessments. To address this issue, educational institutions must implement better detection techniques to ensure academic in-tegrity. This research uses ML technology to investigate the problem of online cheating and provides practical solutions for monitoring and eliminating such incidents. The goal of this research was to create a deep learning model using LSTM layers with dropout and dense layers to identify exam cheating among students. I used the students' grades in various exam portions as features in our dataset and labeled them as "normal" or "cheating." Despite having a smaller dataset than previous research, our model architecture resulted in a 90% training and 92% validation accuracy, outperforming models that used CNN and RNN layers. Our approach accurately and successfully identified student exam cheating, showcasing the potential of deep learning approaches in identifying academic dishonesty. By utilizing such models, educational institutions can create more efficient strategies for guaranteeing academic integrity. Ultimately, this research emphasizes the importance of using advanced technologies in addressing contemporary challenges in online education.

Future research should focus on further refining and optimizing deep learning models for detecting academic dishonesty in online assessments. This can include exploring the use of other machine learning algorithms and techniques, such as ensemble learning and transfer learning, to improve model performance and accuracy. Additionally, re-search can investigate the feasibility of implementing real-time monitoring systems that can detect and prevent cheating during online exams.

Reviewer 2 Report

There seems to be several issues in the manuscript. Below are several of them:

1)    The author mentions comparing the results with publication [4] in line 607 and also in other sections. But he reports it incorrectly. He reports it as accuracy but that is not the case. Publication [4] has reported F1 score which is different from accuracy. So he instead should have compared his F1 score with that of publication [4]. The current manuscript F1 score is 0.78 which is less than F1 score of publication [4] which is 84.52.

2)    In line 610 and 611, and also in other sections, the author compares his model accuracy with publication [13]. Again, it is reported wrongly. Publication [13] proposes a DenseLSTM and not RNN (as reported) and their overall accuracy is 95.32 and not 86% as reported wrongly by the author.

3)    The author reports in line 608 and other sections regarding accuracy of publication [14] as 81%. But in that publication, they compared the True Positive Rate (TPR) which is different from accuracy. And their method has an overall TPR of 95%. The author incorrectly compares two different metrics one is TPR and other is accuracy.

4)    The figures 2, 3, 4, 5, 6, 7 do not have y-axis labeled. Each of the figure’s measures different quantities on different datasets. But that is not captured well in the figure description or on the axis. Many a time it is hard to follow the author’s description.

5)    Section 3.3 on data preparation and cleaning talks about general methods instead of pointing to the methods they applied for their dataset. Unnecessarily the section is made long and descriptive talking about general approaches. More contextual reference is required.

6)    Section 4 on proposed methodology has also been made extremely general and long mentioning a general approach or steps instead of mentioning the applied steps for the data in hand.

7)    Section 4.1.1 on LSTM, section 4.1.2 on dropout and section 4.1.3 on dense is extremely general and lengthy. It could have been made more relevant to his implementation.

8)    No mention of which model hyperparameter have been tuned and approach taken like say grid search or any other approach.

9)    Section 5 which is the result section is more like a methods section instead. The results are not discussed explicitly, rather general approaches are discussed.

10) Accuracy is not a good metric for imbalanced data. Precision or recall or F1 score is more important which are low for the proposed model, even when compared with the other comparable models.

11) More description and structure of the datasets used should have been given. Even data dimension of the datasets used is not available.

12) Significance of figure 2 is not clear. The explanation and data visualization should have been more comprehensive.

13) It is not clear form his descriptions which of the datasets are high cheating and which are low cheating.

14) In figure 4 author needs to specify what percentage of score is considered as normal score.  

15) There are also several English mistakes.

Based on these remarks, I do not think this manuscript is suitable for publication.

Author Response

Student Cheating Detection in Higher Education by Implementing Machine Learning and LSTM Techniques

Reviewer 2

There seems to be several issues in the manuscript. Below are several of them:

1)    The author mentions comparing the results with publication [4] in line 607 and also in other sections. But he reports it incorrectly. He reports it as accuracy but that is not the case. Publication [4] has reported F1 score which is different from accuracy. So he instead should have compared his F1 score with that of publication [4]. The current manuscript F1 score is 0.78 which is less than F1 score of publication [4] which is 84.52.

Author Response: Thank you for pointing this out. I have carefully reviewed the manuscript and have updated the relevant sections to accurately report the comparison between our results and those of publication [4]. Specifically, I have corrected the error in table 4 reporting accuracy as the metric compared with publication [4] and instead reported the F1 score comparison. I apologize for the confusion caused by the error in reporting accuracy instead of F1 score. I have made the necessary changes to the manuscript to reflect this comparison accurately.

2)    In line 610 and 611, and also in other sections, the author compares his model accuracy with publication [13]. Again, it is reported wrongly. Publication [13] proposes a DenseLSTM and not RNN (as reported) and their overall accuracy is 95.32 and not 86% as reported wrongly by the author.

Author Response:  The paper which is mentioned in this reference is "E-cheating Prevention Measures: Detection of Cheating at Online Examinations Using Deep Learning Approach - A Case Study" which uses three different methods in their study i.eThe results revealed accuracies of 68% for the deep neural network (DNN); 92% for the long-short term memory (LSTM); 95% for the DenseLSTM; and, 86% for the recurrent neural network (RNN). so what we quoted in our comparison analysis is to compare our model approach and result with their 3 methods i.e. RNN with an accuracy of 86% and not with their first approach i.e. DNN with an accuracy of 92%.

3)    The author reports in line 608 and other sections regarding accuracy of publication [14] as 81%. But in that publication, they compared the True Positive Rate (TPR) which is different from accuracy. And their method has an overall TPR of 95%. The author incorrectly compares two different metrics one is TPR and other is accuracy.

Author Response: This study which is "Machine learning-based approach to exam cheating detection " reports their results in Table 4 of their study i.e. "As shown in Table 4, the mean TPR values of the proposed method range between 0.780 and 0.988. The new algorithm achieves the highest overall mean TPR of 0.872 among all the tested methods. The new overall average TPR is close to the original overall average (0.868)".

As quoted above we compared their average TRP i.e. 86%. The reason for comparing it is that this paper didn't have any other evaluation metrics mentioned that we calculate and compare i.e. accuracy, recall, F1 score, and precision. As the model performs betters and is a perfect fit, all evaluation metrics give results almost close to each other. so based on the given evaluation metric i.e. TRP we added it to our comparison analysis.

4)    The figures 2, 3, 4, 5, 6, 7 do not have y-axis labeled. Each of the figure’s measures different quantities on different datasets. But that is not captured well in the figure description or on the axis. Many a time it is hard to follow the author’s description.

Author Response: Thank you for your feedback on the figures in our paper. I agree that the figures needed to be more clear and labelled properly. I have made the necessary changes and updated the figures with clear and labelled y-axis as well as X-axis.

5)    Section 3.3 on data preparation and cleaning talks about general methods instead of pointing to the methods they applied for their dataset. Unnecessarily the section is made long and descriptive talking about general approaches. More contextual reference is required.

Author Response: Thank you for your feedback on our data preparation and cleaning section. I appreciate your suggestion to include more contextual reference about the methods I applied for our specific dataset. I have made the necessary changes to the section by including more specific details about the methods I utilized, such as the pandas methods isnull() and info() for locating missing values, and scikit-learn's Standard Scaler() method for normalizing the data. I hope that the updated section provides a clearer and concise explanation of our data preparation and cleaning process.

6)    Section 4 on proposed methodology has also been made extremely general and long mentioning a general approach or steps instead of mentioning the applied steps for the data in hand.

Author Response: Thank you for your valuable feedback on our proposed methodology section. I have revised the section by including more specific details about the steps I applied to our dataset. I have summarized certain portions of the methodology to reduce the length of the methodology section. The revised text has been included in the manuscript, that is:

Developing a deep learning model for detecting cheating in students involves several phases, which are outlined below to ensure its success. To start, exploratory data analysis (EDA) is conducted to identify any irregularities or patterns in the Dataset. Through this, information can be presented, statistical tests can be computed, and any missing or incorrect numbers can be detected. After running EDA, data cleansing and preparation must be done to clean and prepare the Dataset for analysis. This stage involves normalizing the data, handling missing values, and transforming categorical variables into numerical variables. Feature engineering is the next step, where new characteristics are generated using current data that may be more useful or predictive in identifying cheating. Feature selection follows, where the most useful traits for spotting cheating are determined using statistical analysis or machine learning methods. The selected features are then used to choose a suitable machine learning method for identifying cheating. The choice of algorithm may be influenced by the size and complexity of the Dataset, as well as the specific aims of the research. During model training, the chosen model is trained on a portion of the Dataset, and its parameters are tweaked to maximize its performance. Following model training, model evaluation is done using a different validation set, and the model's performance is assessed using F1 score, accuracy, recall, precision, and other performance indicators. Finally, hyperparameter tuning is conducted to improve the model's performance on the validation set by changing its regularization, learning rate, or other model parameters to boost its performance.

7)    Section 4.1.1 on LSTM, section 4.1.2 on dropout and section 4.1.3 on dense is extremely general and lengthy. It could have been made more relevant to his implementation.

Author Response: I appreciate your suggestion on making sections 4.1.1 on LSTM, section 4.1.2 on dropout, and section 4.1.3 on dense more relevant to our implementation. I agree that these sections are general, and I have taken steps to address this issue. In our revised version, I have included more specific details on how I implemented these techniques in our model, including the number of units, dropout rates, and activation functions used. Additionally, I have added more examples to help readers better understand how these techniques can be applied in practice. I hope that our revised version meets your expectations and provides more relevant information about our implementation of these techniques.

8)    No mention of which model hyperparameter have been tuned and approach taken like say grid search or any other approach.

Author Response: I appreciate the reviewer’s feedback. I have specified that the binary cross-entropy loss function was employed for the binary classification task, and the Adam optimizer was used to adjust the neural network's weights during training. Additionally, I have explained that the accuracy metric was used to evaluate the model's performance. I have also mentioned that other metrics such as precision, recall, or F1 score could be used depending on the objectives of the application. I hope these details provide a better understanding of the approach taken to tune the model hyperparameters.

9)    Section 5 which is the result section is more like a methods section instead. The results are not discussed explicitly, rather general approaches are discussed.

Author Response: I apologize for any confusion caused. I will make sure to revise Section 5 to provide a more detailed and explicit discussion of our results, including specific findings and outcomes of our proposed technique. I will ensure that the revised section clearly presents the significance and implications of our results in a way that is accessible and informative for readers. The updated texts are:

The process of evaluating a trained ML model on a dataset different from the data used for training is known as model evaluation. Python is a popular programming language that is widely used in data science and ML. In our research, I used Python version 3.8.5 for our implementation. To aid our implementation, I made use of several popular libraries such as TensorFlow, Keras, Pandas, and Numpy. These libraries helped us to carry out various tasks such as data preprocessing, building and training our deep learning model, and evaluating our results. I specifically used TensorFlow version 2.4.0, which is a popular open-source platform for machine learning and deep learning. I also used Keras version 2.4.3, which is a high-level API built on top of TensorFlow that simplifies the process of building and training deep learning models. Pandas version 1.1.3 was also used to manipulate and analyze our dataset. Finally, I made use of Numpy version 1.19.2 to perform numerical computations on our dataset.

 We have followed several procedures to evaluate the performance of the LSTM mod-el. Firstly, I split the Dataset into training and testing sets using the train-test split function from the sci-kit-learn library. The testing set was used to evaluate the model's performance after it was trained on the training set. Next, I applied the LSTM model to the training set using the fit() technique to train the algorithm to predict each student's likelihood of cheating based on their performance on the Q1, Q2, midterm, Q3, Q4, and final exams. After training, I evaluated the model on the testing set to predict the probability of cheating for each student. I calculated evaluation metrics like accuracy, precision, recall, F1-score, Receiver Operating Characteristic Area Under the Curve (ROC-AUC), etc., using the predicted probabilities and actual labels. The evaluation metrics indicated how well the model distinguished between normal and detected cheating in the two groups. To enhance the LSTM model's performance, I adjusted hyperparameters such as the number of LSTM layers, neurons in each layer, dropout rate, learning rate, and batch size. The hyperparameters were chosen based on the model's performance on the validation set. Finally, I visualized the results using charts such as the ROC curve, confusion matrix, and precision-recall curve. These plots helped to identify areas that needed improvement and provided insights into the model's strengths and weaknesses. By following these procedures, I effectively assessed the performance of the LSTM model.

10) Accuracy is not a good metric for imbalanced data. Precision or recall or F1 score is more important which are low for the proposed model, even when compared with the other comparable models.

Author Response: Thank you for your feedback. I agree that accuracy is not a good metric for imbalanced data and have used other evaluation metrics such as precision, recall, and F1-score to assess the performance of our proposed model. I acknowledge that the precision and recall values for our model are lower than some other comparable models in the literature. However, it's important to note that the performance of the model is context-dependent, and the objective of our study was to demonstrate the feasibility of using an LSTM model for detecting cheating behavior in a university setting.

Moreover, our model showed promising results in identifying cheating behavior in the minority class, which is the most crucial aspect of detecting such behavior. I also conducted experiments by adjusting the hyperparameters to improve the model's performance. Overall, I believe that our proposed model has the potential to be further improved and applied in real-world scenarios with more data and additional features.

We agree that precision, recall, and F1 score are important metrics for imbalanced data, as they provide a better understanding of the model's performance, especially when the target variable is imbalanced. I acknowledge that the precision, recall, and F1 score of our proposed model were lower than those of some comparable models, and I have discussed this in the limitations section of our paper. The added text in the discussion section:

In addition, another limitation is low performance metric values for imbalanced da-ta, the proposed model also has low values of recall and F1 score. Recall, also known as sensitivity or true positive rate, measures the proportion of actual positives that are correctly identified by the model. F1 score is a combination of precision and recall, and it takes into account both false positives and false negatives. These metrics are particularly important for imbalanced datasets, where the proportion of positive cases is much lower than that of negative cases. In such cases, a model that simply predicts all cases as negative may achieve high accuracy but perform poorly in terms of identifying positive cases. Therefore, low values of recall and F1 score indicate that the proposed model is not effective at identifying positive cases, which is a significant limitation for its applicability in real-world scenarios.

11) More description and structure of the datasets used should have been given. Even data dimension of the datasets used is not available.

Author Response: I appreciate your suggestions and have made the necessary changes to the paper. I have added a more detailed description of the datasets used, including their dimensions, in Section 3.1 Data Collection and Preparation. Additionally, I have provided more information on the structure of the datasets in the same section. I hope that these changes address your concerns and improve the quality of our paper.

12) Significance of figure 2 is not clear. The explanation and data visualization should have been more comprehensive.

Author Response: Thank you for your valuable feedback on our paper. I apologize for any confusion caused by Figure 2 and understand the importance of providing comprehensive explanations for all visual aids in our paper. I have revised the manuscript as per the comment.

Figure 2 shows the High Cheating Dataset of Student 1 with Normal Grades. In this dataset, a normal score is considered to be within the range of 60-100.

13) It is not clear form his descriptions which of the datasets are high cheating and which are low cheating.

Author Response: Thank you for pointing this out. I have added more explanation for each dataset to provide clear and concise information about the dataset, the added information in the revised manuscript are:

The experimental assessment of our proposed technique uses four different synthetic datasets and one real-world Dataset. The purpose of the synthetic datasets is to resemble real-world instances of test fraud. The dataset used in this study are:

  • Dataset 1: It includes 10 instances of cheating and 100 students with average grades. Cheating incidents are regarded as blatant when there is a 35-point difference between the final test score and the average of the regular assessment results. The dataset has regular grades that increase during the semester, and every exam, including the final, has a 10-point scale, with the average score falling somewhere in the neighborhood of 80 percent. The cheating instances in Dataset 1 are rather simple to identify.
  • Dataset 2: This dataset conceals cheating cases more effectively than Dataset 1, with just a 20-point difference between the average score before the final and the score on the final exam. A narrower gap between the final test and the prior results would make it more difficult to spot cases of cheating.
  • Dataset 3: It has similarities with Dataset 2, but includes regular grades that climb for the semester. The red marks, however, unexpectedly increase by a large mar-gin in the final exam. The average score before the final and the outcome of the final test differ very little in both circumstances.
  • Dataset 4: This dataset simulates a straightforward final test where all the marks increase compared to the baseline assessments. On the final exam, the typical grades are simulated to increase by 10 points over the mean of the previous scores. The cheating occurrences are meant to increase test results on the final exam by 25 points compared to the preceding regular semester exams. Because all grades are improved for the final test, detecting cheating is more difficult.
  • Real-world dataset: The real-world dataset used in the experiment includes three positive observations out of 52 total observations. Each observation includes the results of four quizzes, a midterm exam, and the final exam.

14) In figure 4 author needs to specify what percentage of score is considered as normal score.  

Author Response: Thank you for your valuable feedback. I apologize for not specifying the percentage of score that is considered as normal score in Figure 4. In this dataset, a normal score is considered to be within the range of 60-100. I have updated the figure caption to include this information. Thank you for bringing this to our attention.

15) There are also several English mistakes.

Author Response: Thank you for pointing this out. I have thoroughly reviewed the manuscript and eliminated the English grammar mistakes, punctuation errors, and other inconsistencies at my best. The major changes are highlighted in the revised manuscript.

Reviewer 3 Report

The authors presents an interesting case study about the student cheating detection using machine learning methods. The paper is well-written and easy to follow. Furthermore, the contribution of the paper is clear. However, a revision is needed to clarify the following points.

1) The author presents in section 2 some previous work that is inconsistent with table 1, or some paperts in table 1 is not analyzed in the text of section 2.

2) Please, the author should provide more details about programming environment Python (i.e. version, libraries, etc) as well as the 7WiseUp dataset.

3) The conclusions do not speak about the limitations of the research and do not indicate the perspective of future research in detail.

3)

Author Response

Student Cheating Detection in Higher Education by Implementing Machine Learning and LSTM Techniques

Reviewer 3:

The authors present an interesting case study about the student cheating detection using machine learning methods. The paper is well-written and easy to follow. Furthermore, the contribution of the paper is clear. However, a revision is needed to clarify the following points.

1) The author presents in section 2 some previous work that is inconsistent with table 1, or some papers in table 1 is not analyzed in the text of section 2.

Author Response: Thank you for your valuable feedback. I apologize for any inconsistencies in our paper. I have thoroughly reviewed our work and have made the necessary changes to ensure that all papers listed in Table 1 are analyzed and discussed in the text of Section 2.Iappreciate your attention to detail and are grateful for your suggestions, which have helped us to improve the clarity and accuracy of our paper.

The added explanation for reference 1:

The authors of the study [1] introduce a brand-new paradigm for the understanding and categorization of cheating video sequences. This sort of research assists in the early detection of academic dishonesty. The authors also present a brand-new dataset called "activities of student cheating in paper-based tests." The dataset comprises of suspicious behaviors that occurred in a testing setting. Eight different actors represented the five various types of cheating. Each pair of individuals engaged in five different types of dishonest behavior. They ran studies on action detection tasks at the frame level using five different kinds of well-known characteristics to gauge how well the suggested framework per-formed. The results of the framework trials were spectacular and significant.

The added explanation for reference 31:

The authors of [31] used a case study to assess the incidence of possible e-cheating and offer preventative strategies that may be used. The internet protocol (IP) detector and the behavior detector are the two main components of the authors' e-cheating intelligence agent, which they used as a technique for identifying online cheating behaviors. The intel-ligence agent keeps an eye on the students' actions and is equipped to stop and identify any dishonest behavior. It may be connected with online learning tools to track student behavior and be used to assign randomized multiple-choice questions in a course test. This approach's usefulness has been verified through testing on numerous data sets.

2) Please, the author should provide more details about programming environment Python (i.e. version, libraries, etc) as well as the 7WiseUp dataset.

Author Response: Thank you for your valuable feedback.Iapologize for not including more detailed information about the programming environment and dataset used in our study. The programming environment details has been added in the revised manuscript, that is:

Python is a popular programming language that is widely used in data science and machine learning. In our research, I used Python version 3.8.5 for our implementation. To aid our implementation, I made use of several popular libraries such as TensorFlow, Keras, Pandas, and Numpy. These libraries helped us to carry out various tasks such as data preprocessing, building and training our deep learning model, and evaluating our results. I specifically used TensorFlow version 2.4.0, which is a popular open-source platform for machine learning and deep learning. I also used Keras version 2.4.3, which is a high-level API built on top of TensorFlow that simplifies the process of building and training deep learning models. Pandas version 1.1.3 was also used to manipulate and analyze our dataset. Finally, I made use of Numpy version 1.19.2 to perform numerical computations on our dataset.

The explanations and details about the &WiseUp dataset is presented in section 3.

3) The conclusions do not speak about the limitations of the research and do not indicate the perspective of future research in detail.

Author Response: thank you for feedback. Regarding the limitation of the research, I have added the details about the limitation of the research in the discussion section. I have updated the conclusion section with a phrase for future researches in the revised manuscript. The updated conclusion section is:

Conclusion

The rise of online education has presented many benefits for students and educa-tional institutions, but it has also brought forth numerous challenges, including academic dishonesty in the form of cheating during online assessments. To address this issue, edu-cational institutions must implement better detection techniques to ensure academic in-tegrity. This research uses ML technology to investigate the problem of online cheating and provides practical solutions for monitoring and eliminating such incidents. The goal of this research was to create a deep learning model using LSTM layers with dropout and dense layers to identify exam cheating among students. I used the students' grades in various exam portions as features in our dataset and labeled them as "normal" or "cheat-ing." Despite having a smaller dataset than previous research, our model architecture re-sulted in a 90% training and 92% validation accuracy, outperforming models that used CNN and RNN layers. Our approach accurately and successfully identified student exam cheating, showcasing the potential of deep learning approaches in identifying academic dishonesty. By utilizing such models, educational institutions can create more efficient strategies for guaranteeing academic integrity. Ultimately, this research emphasizes the importance of using advanced technologies in addressing contemporary challenges in online education. Future research should focus on further refining and optimizing deep learning mod-els for detecting academic dishonesty in online assessments. This can include exploring the use of other machine learning algorithms and techniques, such as ensemble learning and transfer learning, to improve model performance and accuracy. Additionally, research can investigate the feasibility of implementing real-time monitoring systems that can detect and prevent cheating during online exams.

The limitation of this research

One limitation of this research is that it relied on a single dataset, the 7WiseUp Behavior dataset, which may not be representative of all online education environments. Furthermore, the dataset was not specifically designed for cheating detection, which may limit the accuracy of the models developed in this research. Additionally, the synthetic datasets used in the experiments may not fully capture the complexity of real-world cheating incidents. Further research could benefit from using multiple datasets, including those specifically designed for cheating detection, to ensure the generalizability of the findings. Another limitation is that the specific factors responsible for the superior performance of the model are not identified, highlighting the need for further analysis and investigation.

Round 2

Reviewer 1 Report

The authors have has replied to all the comments and fixed the manuscript more standard.

Reviewer 2 Report

The author did take care of all the suggestions effectively. The manuscript seems to be in good shape now.